# Learning Universal Policies via Text-Guided Video Generation

**Yilun Du** [*†‡], **Mengjiao Yang** [*‡§], **Bo Dai** [‡¶], **Hanjun Dai** [‡], **Ofir Nachum** [‡],
**Joshua B. Tenenbaum** [†], **Dale Schuurmans** [‡∥], **Pieter Abbeel** [§]

MIT[†]     Google DeepMind[‡]     UC Berkeley[§]     Georgia Tech[¶]     University of Alberta[∥]

https://universal-policy.github.io/

## Abstract

A goal of artificial intelligence is to construct an agent that can solve a wide variety of tasks. Recent progress in text-guided image synthesis has yielded models with an impressive ability to generate complex novel images, exhibiting combinatorial generalization across domains. Motivated by this success, we investigate whether such tools can be used to construct more general-purpose agents. Specifically, we cast the sequential decision making problem as a text-conditioned video generation problem, where, given a text-encoded specification of a desired goal, a planner synthesizes a set of future frames depicting its planned actions in the future, after which control actions are extracted from the generated video. By leveraging text as the underlying goal specification, we are able to naturally and combinatorially generalize to novel goals. The proposed policy-as-video formulation can further represent environments with different state and action spaces in a unified space of images, which, for example, enables learning and generalization across a variety of robot manipulation tasks. Finally, by leveraging pretrained language embeddings and widely available videos from the internet, the approach enables knowledge transfer through predicting highly realistic video plans for real robots[2].

## 1 Introduction

Building models that solve a diverse set of tasks has become a dominant paradigm in the domains of vision and language. In natural language processing, large pretrained models have demonstrated remarkable zero-shot learning of new language tasks [1, 2, 3]. Similarly, in computer vision, models such as those proposed in [4, 5] have shown remarkable zero-shot classification and object recognition capabilities. A natural next step is to use such tools to construct agents that can complete different decision making tasks across many environments.

However, training such agents faces the inherent challenge of environmental diversity, since different environments operate with distinct state action spaces (e.g., the joint space and continuous controls in MuJoCo are fundamentally different from the image space and discrete actions in Atari). Such diversity hampers knowledge sharing, learning, and generalization across tasks and environments. Although substantial effort has been devoted to encoding different environments with universal tokens in a sequence modeling framework [6], it is unclear whether such an approach can preserve the rich knowledge embedded in pretrained vision and language models and leverage this knowledge to transfer to downstream reinforcement learning (RL) tasks. Furthermore, it is difficult to construct reward functions that specify different tasks across environments.

In this work, we address the challenges in environment diversity and reward specification by leveraging *video* (i.e., image sequences) as a universal interface for conveying action and observation

---

[*] denotes equal contribution. Correspondence to yilundu@mit.edu and sherryy@berkeley.edu.

[2]See video visualizations at https://universal-policy.github.io.

37th Conference on Neural Information Processing Systems (NeurIPS 2023).

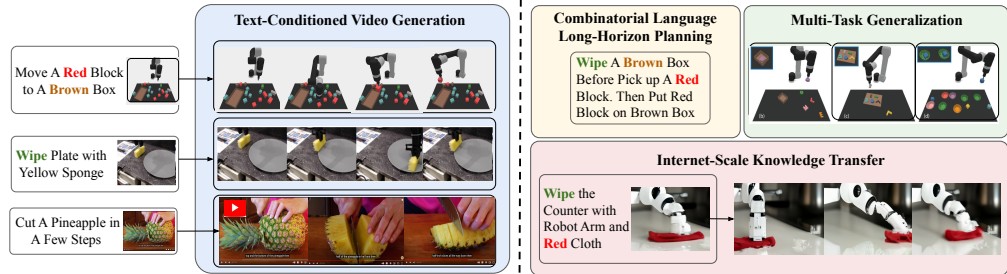

Figure 1: **Text-Conditional Video Generation as Universal Policies.** Text-conditional video generations enables us to train general purpose policies on wide sources of data (simulated, real robots and YouTube) which may be applied to downstream multi-task settings requiring combinatorial language generalization, long-horizon planning, or internet-scale knowledge.

behavior in different environments, and *text* as a universal interface for expressing task descriptions. In particular, we design a video generator as a planner that sequentially conditions on a current image frame and a text passage describing a current goal (i.e., the next high-level step) to generate a trajectory in the form of an image sequence, after which an inverse dynamics model is used to extract the underlying actions from the generated video. Such an approach allows the universal nature of language and video to be leveraged in generalizing to novel goals and tasks across diverse environments. Specifically, we instantiate the text-conditioned video generation model using video diffusion. A set of underlying actions are then regressed from the synthesized frames and used to construct a policy to implement the planned trajectory. Since the language description of a task is often highly correlated with control actions, text-conditioned video generation naturally focuses generation on action relevant parts of the video. The proposed model, *UniPi*, is visualized in Figure 1.

We have found that formulating policy generation via text-conditioned video synthesis yields the following advantages:

**Combinatorial Generalization.**    The rich combinatorial nature of language can be leveraged to synthesize novel combinatorial behaviors in the environment. This enables the proposed approach to rearrange objects to new unseen combinations of geometric relations, as shown in Section 4.1.

**Multi-task Learning.**    Formulating action prediction as a video prediction problem readily enables learning across many different tasks. We illustrate in Section 4.2 how this enables learning across language-conditioned tasks and generalizing to new ones at test time without finetuning.

**Action Planning.**    The video generation procedure corresponds to a planning procedure where a sequence of frames representing actions is generated to reach the target goal. Such a planning procedure is naturally hierarchical: a temporally sparse sequence of images toward a goal can first be generated, before being refined with a more specific plan. Moreover, the planning procedure is steerable, in the sense that the plan generation can be biased by new constraints introduced at test-time through test-time sampling. Finally, plans are produced in a video space that is naturally interpretable by humans, making action verification and plan diagnosis easy. We illustrate the efficacy of hierarchical sampling in Table 2 and steerability in Figure 7.

**Internet-Scale Knowledge Transfer.**    By pretraining a video generation model on a large-scale text-video dataset recovered from the internet, one can recover a vast repository of "demonstrations" that aid the construction of a text-conditioned policy in novel environments. We illustrate how this enables the realistic synthesis of robot motion videos from language instructions in Section 4.3.

The main contribution of this work is to formulate text-conditioned video generation as a universal planning strategy from which diverse behaviors can be synthesized. While such an approach departs from typical policy generation in RL, where subsequent actions to execute are directly predicted from a current state, we illustrate that UniPi exhibits notable generalization advantages over traditional policy generation methods across a variety of domains.

## 2  Problem Formulation

We first motivate a new abstraction, the *Unified Predictive Decision Process (UPDP)*, as an alternative to the Markov Decision Process (MDP) commonly used in RL, and then show an instantiation of a UPDP with diffusion models.

## 2.1 Markov Decision Process

The Markov Decision Process [7] is a broad abstraction used to formulate many sequential decision making problems. Many RL algorithms have been derived from MDPs with empirical success [8, 9, 10], but existing algorithms are typically unable to combinatorially generalize across different environments. Such difficulty can be traced back to certain aspects of the underlying MDP abstraction:

**i)** The lack of a universal state interface across different control environments. In fact, since different environments typically have separate underlying state spaces, one would need to a construct a complex state representation to represent all environments, making learning difficult.

**ii)** The explicit requirement of a real-valued reward function in an MDP. The RL problem is usually defined as maximizing the accumulated reward in an MDP. However, in many practical applications, how to design and transfer rewards is unclear and different across environments.

**iii)** The dynamics model in an MDP is environment and agent dependent. Specifically, the dynamics model $T(s'|s, a)$ that characterizes the transition between states $(s, s')$ under action $a$ is explicitly dependent to the environment and action space of the agent, which can be significantly different between different agents and tasks.

## 2.2 Unified Predictive Decision Process

These difficulties inspire us to construct an alternative abstraction for unified sequential decision making across diverse environments. Our abstraction, termed *Unified Predictive Decision Process (UPDP)*, exploits images as a universal interface across environments, texts as task specifiers to avoid reward design, and a task-agnostic planning module separated from environment-dependent control to enable knowledge sharing and generalization.

Formally, we define a UPDP to be a tuple $\mathcal{G} = \langle \mathcal{X}, \mathcal{C}, H, \rho \rangle$, where $\mathcal{X}$ denotes the observation space of images, $\mathcal{C}$ denotes the space of textual task descriptions, $H \in \mathcal{N}$ is a finite horizon length, and $\rho(\cdot|x_0, c) : \mathcal{X} \times \mathcal{C} \to \Delta(\mathcal{X}^H)$ is a conditional video generator. That is, $\rho(\cdot|x_o, c) \in \Delta(\mathcal{X}^H)$ is a conditional distribution over $H$-step image sequences determined by the first frame $x_0$ and the task description $c$. Intuitively, $\rho$ synthesizes $H$-step image trajectories that illustrate possible paths for completing a target task $c$. For simplicity, we focus on finite horizon, episodic tasks.

Given a UPDP $\mathcal{G}$, we define a trajectory-task conditioned policy $\pi(\cdot|\{x_h\}_{h=0}^H, c) : \mathcal{X}^{H+1} \times \mathcal{C} \to \Delta(\mathcal{A}^H)$ to be a conditional distribution over $H$-step action sequences $\mathcal{A}^H$. Ideally, $\pi(\cdot|\{x_h\}_{h=0}^H, c)$ specifies a conditional distribution of action sequences that achieves the given trajectory $\{x_h\}_{h=0}^H$ in the UPDP $\mathcal{G}$ for the given task $c$. To achieve such an alignment, we will consider an offline RL scenario where we have access to a dataset of existing experience $\mathcal{D} = \{(x_i, a_i)_{i=0}^{H-1}, x_H, c\}_{j=1}^n$ from which both $\rho(\cdot|x_0, c)$ and $\pi(\cdot|\{x_h\}_{h=0}^H, c)$ can be estimated.

In contrast to an MDP, a UPDP directly models video-based trajectories and bypasses the need to specify a reward function beyond the textual task description. Since the space of video observations $\mathcal{X}^H$ and task descriptions $\mathcal{C}$ are both naturally shared across environments and easily interpretable by humans, any video-based planner $\rho(\cdot|x_0, c)$ can be conveniently *reused*, *transferred* and *debugged*. Another benefit of a UPDP over an MDP is that UPDP isolates the video-based planning with $\rho(\cdot|x_0, c)$ from the deferred action selection using $\pi(\cdot|\{x_h\}_{h=0}^H, c)$. This design choice isolates planning decisions from action-specific mechanisms, allowing the planner to be environment and agent agnostic.

UPDP can be understood as implicitly planning over an MDP and directly outputting an optimal trajectory based on the given instructions. Such a UPDP abstraction bypasses reward design, state extraction and explicit planning, and allows for *non-Markovian* modeling of an image-based state space. However, learning a planner in UPDP requires videos and task descriptions, whereas traditional MDPs do not require such data, so whether an MDP or UPDP is more suitable for a given task depends on what types of training data are available. Although the non-Markovian model and the requirement of video and text data induce additional difficulties in UPDP comparing to MDP, it is possible to leverage existing large text-video models that have been pretrained on massive, web-scale datasets to alleviate these complexities.

## 2.3 Diffusion Models for UPDP

Let $\tau = [x_1, \ldots, x_H] \in \mathcal{X}^H$ denote a sequence of images. We leverage the significant recent advances in diffusion models for capturing the conditional distribution $\rho(\tau|x_0, c)$, which we will

leverage as a text and initial-frame conditioned video generator in a UPDP. We emphasize that the UPDP formulation is also compatible with other probabilistic models, such as a variational autoencoder [11], energy-based model [12, 13], or generative adversarial network [14]. For completeness we briefly cover the core formulation at a high-level, but defer details to background references [15].

We start with an unconditional model. A continuous-time diffusion model defines a forward process $q_k(\tau_k|\tau) = \mathcal{N}(\cdot; \alpha_k\tau, \sigma_k^2 I)$, where $k \in [0,1]$ and $\alpha_k, \sigma_k^2$ are scalars with predefined schedules. A generative process $p(\tau)$ is also defined, which reverses the forward process by learning a denoising model $s(\tau_k, k)$. Correspondingly $\tau$ can be generated by simulating this reverse process with an ancestral sampler [16] or numerical integration [17]. In our case, the unconditional model needs to be further adapted to condition on both the text instruction $c$ and the initial image $x_0$. Denote the conditional denoiser as $s(\tau_k, k|c, x_0)$. We leverage classifier-free guidance [18] and use $\hat{s}(\tau_k, k|c, x_0) = (1+\omega)s(\tau_k, k|c, x_0) - \omega s(\tau_k, k)$ as the denoiser in the reverse process for sampling, where $\omega$ controls the strength of the text and first-frame conditioning.

## 3   Decision Making with Videos

Next we describe the proposed approach UniPi in detail, which is a concrete instantiation of the diffusion UPDP. UniPi incorporates the two main components discussed in Section 2, as shown in Figure 2: **(i)** a diffusion model for the universal video-based planner $\rho(\cdot|x_0, c)$, which synthesizes videos conditioned on the first frame and task descriptions; and **(ii)** a task-specific action generator $\pi(\cdot|\{x_h\}_{h=0}^H, c)$, which infers actions sequences from generated videos through inverse dynamics modeling.

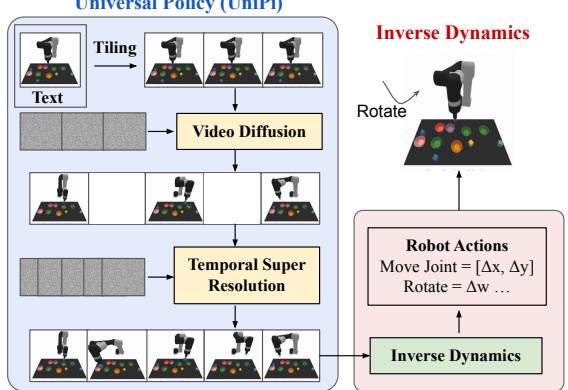

Figure 2: Given an input observation and text instruction, we plan a set of images representing agent behavior. Images are converted to actions using an inverse dynamics model.

### 3.1   Universal Video-Based Planner

Encouraged by the recent success of text-to-video models [19], we seek to construct a video diffusion module as the trajectory planner, which can faithfully synthesize future image frames given an initial frame and textual task description. However, the desired planner departs from the typical setting in text-to-video models [20, 19] which normally generate unconstrained videos given a text description. Planning through video generation is more challenging as it requires models to both be able to generate constrained videos that start at a specified image, and then complete the target task. Moreover, to ensure valid action inference across synthesized frames in a video, the video prediction module needs to be able to track the underlying environment state across synthesized video frames.

**Conditional Video Synthesis.**   To generate a valid and executable plan, a text-to-video model must synthesize a constrained video plan starting at an initial image that depicts the initial configuration of the agent and environment. One approach to solve this problem is to modify the underlying *test-time* sampling procedure of an unconditional model, by fixing the first frame of the generated video plan to always begin at the observed image, as done in [21]. However, we found that this performed poorly and led to subsequent frames in the video plan to deviate significantly from the original observed image. Instead, we found it more effective to explicitly train a constrained video synthesis model by providing the first frame of each video as explicit conditioning context during training.

**Trajectory Consistency through Tiling.**   Existing text-to-video models typically generate videos where the underlying environment state changes significantly during the temporal duration [19]. To construct an accurate trajectory planner, it is important that the environment remain consistent across all time points. To enforce environment consistency in conditional video synthesis, we provide, as additional context, the observed image when denoising each frame in the synthesized video. In particular, we re-purpose a temporal super-resolution video diffusion architecture, and provide as context the conditioned visual observation tiled across time, as the opposed to a low temporal-resolution video for denoising at each timestep. In this model, we directly concatenate each

intermediate noisy frame with the conditioned observed image across sampling steps, which serves as a strong signal to maintain the underlying environment state across time.

**Hierarchical Planning.** When constructing plans in high dimensional environments with long time horizons, directly generating a set of actions to reach a goal state quickly becomes intractable due to the exponential blow-up of the underlying search space. Planning methods often circumvent this issue by leveraging a natural hierarchy in planning. Specifically, planning methods first construct coarse plans operating on low dimensional states and actions, which may then be refined into plans in the underlying state and action spaces. Similar to planning, our conditional video generation procedure likewise exhibits a natural temporal hierarchy. We first generate videos at a coarse level by sparsely sampled videos ("abstractions") of the desired behavior along the time axis. Then we refine the videos to represent valid behavior in the environment by super-resolving videos across time. Meanwhile, coarse-to-fine super-resolution further improves consistency via interpolation between frames.

**Flexible Behavioral Modulation.** When planning a sequence of actions to a given sub-goal, one can readily incorporate external constraints to modulate the generated plan. Such test-time adaptability can be implemented by composing a prior $h(\tau)$ during plan generation to specify desired constraints across the synthesized action trajectory [21], which is also compatible with UniPi. In particular, the prior $h(\tau)$ can be specified using a learned classifier on images to optimize a particular task, or as a Dirac delta on a particular image to guide a plan towards a particular set of states. To train the text-conditioned video generation model, we utilize the video diffusion algorithm in [19], where pretrained language features from T5 [22] are encoded. Please see Appendix A for the underlying architecture and training details.

## 3.2 Task Specific Action Adaptation

Given a set of synthesized videos, we may train a small task-specific inverse-dynamics model to translate frames into a set of actions as described below.

**Inverse Dynamics.** We train a small model to estimate actions given input images. The training of the inverse dynamics is independent from the planner and can be done on a separate, smaller and potentially suboptimal dataset generated by a simulator.

**Action Execution.** Finally, we generate an action sequence given $x_0$ and $c$ by synthesizing $H$ image frames and applying the learned inverse-dynamics model to predict the corresponding $H$ actions. Inferred actions can then be executed via *closed-loop* control, where we generate $H$ new actions after each step of action execution (i.e., model predictive control), or via *open-loop* control, where we sequentially execute each action from the intially inferred action sequence. For computational efficiency, we use an open-loop controller in all our experiments in this paper.

## 4 Experimental Evaluation

The focus of these experiments is to evaluate UniPi in terms of its ability to enable effective, generalizable decision making. In particular, we evaluate

(1) the ability to combinatorially generalize across different subgoals in Section 4.1,
(2) the ability to effectively learn and generalize across many tasks in Section 4.2,
(3) the ability to leverage existing videos on the internet to generalize to complex tasks in Section 4.3.

See experimental details in Appendix A. Additional results are given in Appendix B and videos in the supplement.

## 4.1 Combinatorial Policy Synthesis

First, we measure the ability of UniPi to combinatorially generalize to different language tasks.

**Setup.** To measure combinatorial generalization, we use the combinatorial robot planning tasks in [23]. In this task, a robot must manipulate blocks in an environment to satisfy language instructions, i.e., *put a red block right of a cyan block.* To accomplish this task, the robot must first pick up a white block, place it in the appropriate bowl to paint it a particular color, and then pick up and place the block in a plate so that it satisfies the specified relation. In contrast to [23] which uses

| Model | Seen | | Novel | |
|---|---|---|---|---|
| | **Place** | **Relation** | **Place** | **Relation** |
| State + Transformer BC [24] | $19.4 \pm 3.7$ | $8.2 \pm 2.0$ | $11.9 \pm 4.9$ | $3.7 \pm 2.1$ |
| Image + Transformer BC [24] | $9.4 \pm 2.2$ | $11.9 \pm 1.8$ | $9.7 \pm 4.5$ | $7.3 \pm 2.6$ |
| Image + TT [25] | $17.4 \pm 2.9$ | $12.8 \pm 1.8$ | $13.2 \pm 4.1$ | $9.1 \pm 2.5$ |
| Diffuser [21] | $9.0 \pm 1.2$ | $11.2 \pm 1.0$ | $12.5 \pm 2.4$ | $9.6 \pm 1.7$ |
| UniPi (Ours) | **59.1** $\pm 2.5$ | **53.2** $\pm 2.0$ | **60.1** $\pm 3.9$ | **46.1** $\pm 3.0$ |

Table 1: **Task Completion Accuracy in Combinatorial Environments.** UniPi generalizes to seen and novel combinations of language prompts in Place (e.g., place X in Y) and Relation (e.g., place X to the left of Y) tasks.

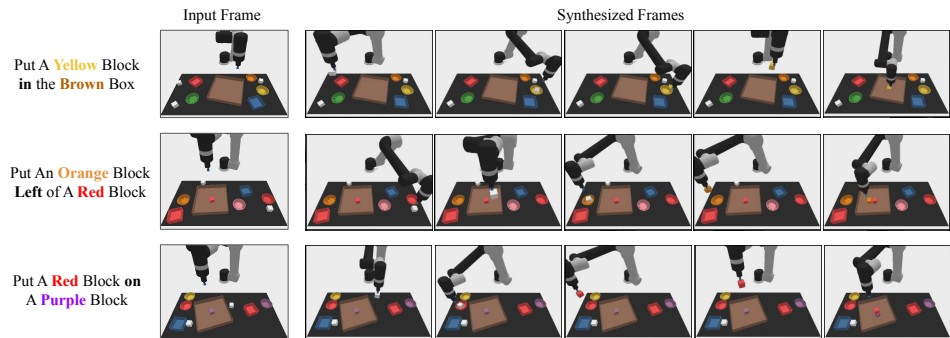

Figure 3: **Combinatorial Video Generation.** Generated videos for unseen language goals at test time.

pre-programmed pick and place primitives for action prediction, we predict actions in the continuous robotic joint space for both the baselines and our approach.

We split the language instructions in this environment into two sets: one set of instructions (70%) that is seen during training, and another set (30%) that is only seen during testing. The precise locations of individual blocks, bowls, and plates in the environment are fully randomized in each environment iteration. We train the video model on 200k example videos of generated language instructions in the train set. Details of this environment can be found in Appendix A. We constructed demonstrations of videos in this task by using a scripted agent.

**Baselines.** We compare the proposed approach with three separate representative approaches. First, we compare to existing work that uses goal-conditioned transformers to learn across multiple environments, where goals can be specified as episode returns [26], expert demonstrations [6], or text and images [24]. To represent these baselines, we construct a transformer behavior cloning (BC) agent to predict the subsequent action to execute given the task description and either the visual observation (Image + Transformer BC) or the underlying robot joint state (State + Transformer BC). Second,

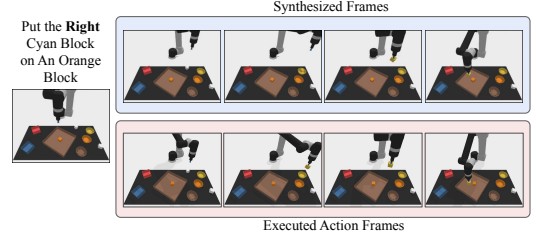

Figure 4: **Action Execution.** Synthesized video plans and executed actions in the simulated environment. The two video plans roughly align with each other.

given that our approach regresses a sequence of actions to execute, we further compare with transformer models that regress a sequence of future actions to execute, similar to the goal-conditioned behavioral cloning of the Trajectory Transformer [25] (Image + TT). Finally, to highlight the importance of the video-as-policy approach, we compare UniPi with learning a diffusion process that, conditioned on an image observation, directly infers future robot actions in the joint space (as opposed to diffusing future image frames), corresponding to [21, 27]. For both our method and each baseline, we condition the policy on encoded language instructions using pretrained T5 embeddings. Note that in this setting, existing offline reinforcement learning baselines are not directly applicable as we do not have access to the reward functions in the environment.

**Metrics.** To compare UniPi with baselines, we measure final task completion accuracy across new instances of the environment and associated language prompts. We subdivide the evaluation along two axes: **(1)** whether the language instruction has been seen during training and **(2)** whether the

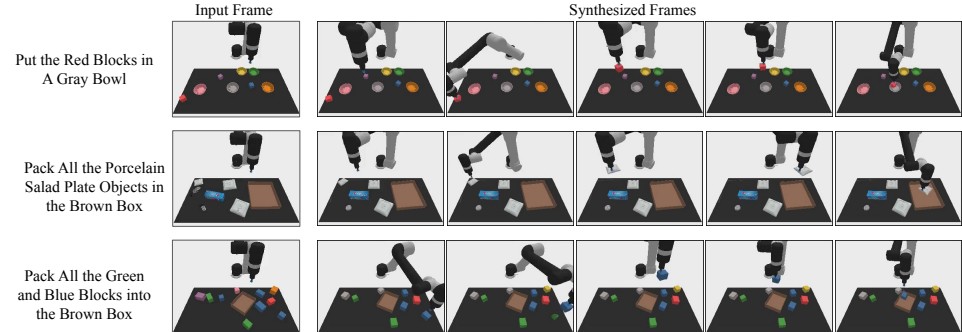

Figure 5: **Multitask Video Generation.** Generated video plans on new test tasks in the multitask setting. UniPi is able to synthesize plan across a set of environments.

| Frame Condition | Frame Consistency | Temporal Heirarchy | Place | Relation |
|---|---|---|---|---|
| No | No | No | $13.2 \pm 3.2$ | $12.4 \pm 2.4$ |
| Yes | No | No | $52.4 \pm 2.9$ | $34.7 \pm 2.6$ |
| Yes | Yes | No | $53.2 \pm 3.0$ | $39.4 \pm 2.8$ |
| Yes | Yes | Yes | $\mathbf{59.1} \pm 2.5$ | $\mathbf{53.2} \pm 2.0$ |

Table 2: **Task Completion Accuracy Ablations.** Each component of UniPi improves its performance. Performance reported on the seen place and relation tasks.

language instruction specifies placing a block in relation to some other block as opposed to direct pick-and-place.

**Combinatorial Generalization.** In Table 1, we find that UniPi generalizes well to both seen and novel combinations of language prompts . We illustrate our action generation pipeline in Figure 4 and different generated video plans using our approach in Figure 3.

**Ablations.** In Table 2, we ablate UniPi on seen language instructions and in-relation-to tasks. Specifically, we study the effect of conditioning the video generative model on the first observation frame (frame condition), tiling the observed frame across timesteps (frame consistency) and super-resolving video generation across time (temporal hierarchy). In settings where frame consistency is not enforced, we provide a zeroed out image as context to the non-start frames in a video. We found that all components of UniPi

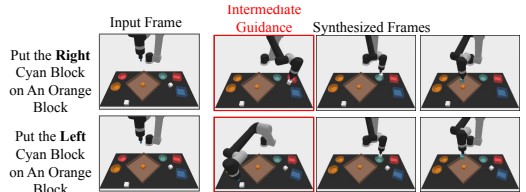

Figure 7: **Adaptable Planning.** By guiding test-time sampling towards a an intermediate image, we can adapt our planning procedure to move a particular block.

are crucial for good performance. We found that frame conditioning and consistency enabled videos to be consistent with the observed image and temporal hierarchy enabled more detailed video plans.

**Adaptability.** We next assess the ability of UniPi to adapt at test time to new constraints. In Figure 7, we illustrate the ability to construct plans which color and move one particular block to a specified geometric relation.

## 4.2 Multi-Environment Transfer

We next evaluate the ability of UniPi to effectively learn across a set of different tasks and generalize, at test time, to a new set of unseen environments.

**Setup.** To measure multi-task learning and transfer, we use the suite of language guided manipulation tasks from [28]. We train our method using demonstrations across a set of 10 separate tasks from [28], and evaluate the ability of our approach to transfer to 3 different test tasks. Using a scripted oracle agent, we generate a set of 200k videos of language execution in the environment. We report the underlying accuracy in which each language instruction is completed. Details of this environment can be found in Appendix A.

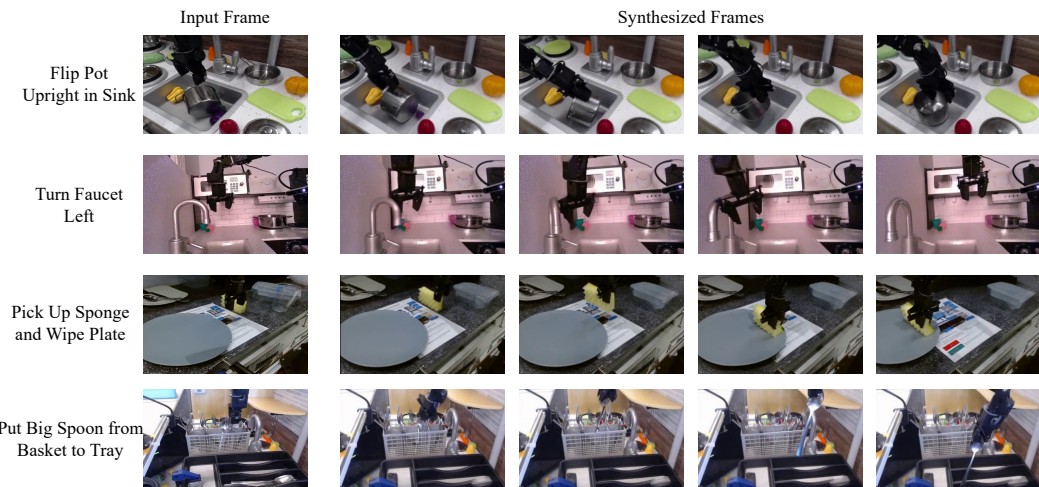

Figure 6: **High Fidelity Plan Generation.** UniPi can generate high resolution video plans across different language prompts.

| Model | Place Bowl | Pack Object | Pack Pair |
|---|---|---|---|
| State + Transformer BC | $9.8 \pm 2.6$ | $21.7 \pm 3.5$ | $1.3 \pm 0.9$ |
| Image + Transformer BC | $5.3 \pm 1.9$ | $5.7 \pm 2.1$ | $7.8 \pm 2.6$ |
| Image + TT | $4.9 \pm 2.1$ | $19.8 \pm 0.4$ | $2.3 \pm 1.6$ |
| Diffuser | $14.8 \pm 2.9$ | $15.9 \pm 2.7$ | $10.5 \pm 2.4$ |
| UniPi (Ours) | $\mathbf{51.6 \pm 3.6}$ | $\mathbf{75.5 \pm 3.1}$ | $\mathbf{45.7 \pm 3.7}$ |

Table 3: **Task Completion Accuracy on Multitask Environment.** UniPi generalizes well to new environments when trained on a set of different multi-task environments.

**Baselines.** We use the same baseline methods as in Section 4.1. While our environment setting is similar to that of [28], this method is not directly comparable to our approach, as CLIPort abstracts actions to the existing primitives of pick and place as opposed to using the joint space of a robot. CLIPort is also designed to solve the significantly simpler problem of inferring only the poses upon which to pick and place objects (with no easy manner to adapt to our setting).

**Multitask Generalization.** In Table 3 we present results of UniPi and baselines across new tasks. The UniPi approach is able to generalize and synthesize new videos and decisions of different language tasks, and can generate videos consisting of picking different kinds of objects and different colored objects. We further present video visualizations of our approach in Figure 5.

## 4.3 Real World Transfer

Finally we evaluate the extent to which UniPi can generalize to real world scenarios and construct complex behaviors by leveraging widely available videos on the internet.

**Setup.** Our training data consists of an internet-scale pretraining dataset and a smaller real-world robotic dataset. The pretraining dataset uses the same data as [19], which consists of 14 million video-text pairs, 60 million image-text pairs, and the publicly available LAION-400M image-text dataset. The robotic dataset is adopted from the Bridge dataset [29] with 7.2k video-text pairs, where we use the task IDs as texts. We partition the 7.2k video-text pairs into train (80%) and test (20%) splits. We pretrain UniPi on the pretraining dataset followed by finetuning on the train split of the Bridge data. Architectural details can be found in Appendix A.

**Video Synthesis.** We are particularly interested in the effect of pretraining on internet-scale video data that is not specific to robotics. We report the CLIP scores, FIDs, and FVDs (averaged across frames and computed on 32 samples) of UniPi trained on Bridge data, with and without pretraining. As shown in Table 4, UniPi with pretraining achieves significantly higher FID and FVD and a marginally better CLIP score than UniPi without pretraining, suggesting that pretraining on non-robot data helps with generating plans for robots. Interestingly, UniPi without pretraining often synthesizes plans that fail to complete the task (Figure 6), which is not well reflected in the CLIP

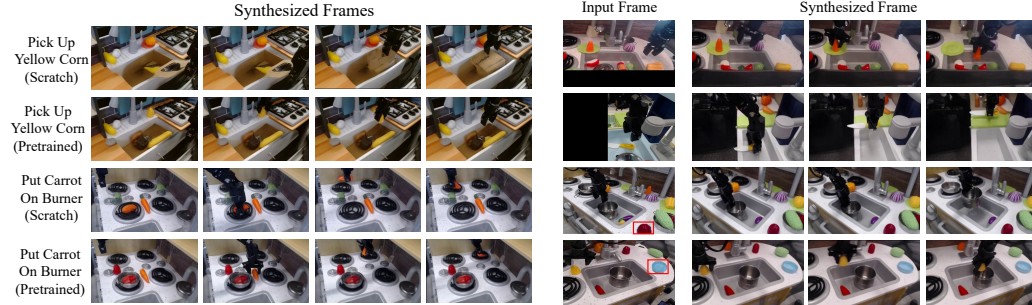

Figure 8: **Pretraining Enables Combinatorial Generalization.** Internet pretraining enables UniPi to synthesize videos of tasks not seen in training. In contrast, a model trained from scratch incorrectly generates plans of different tasks.

Figure 9: **Robustness to Background Change.** UniPi learns to be robust to changes of underlying background, such as black cropping or the addition of photo-shopped objects.

| Model (24x40) | CLIP Score ↑ | FID ↓ | FVD ↓ | Success ↑ |
|---|---|---|---|---|
| No Pretrain | $24.43 \pm 0.04$ | $17.75 \pm 0.56$ | $288.02 \pm 10.45$ | 72.6% |
| Pretrain | $\mathbf{24.54 \pm 0.03}$ | $\mathbf{14.54 \pm 0.57}$ | $\mathbf{264.66 \pm 13.64}$ | **77.1%** |

Table 4: **Video Generation Quality of UniPi on Real Environment.** The use of existing data on the internet improves video plan predictions under all metrics considered.

score, suggesting the need for better generation metrics for control-specific tasks. To tackle the lack of such a metric, we develop a surrogate metric for evaluating task success from the generated videos. Specifically, we train a success classifier that takes in the last frame of a generated video and predicts whether the task is successful or not. We find that training a model from scratch achieves 72.6% success while finetuning from a pretrained model improves performance to 77.1%. In both settings, generated videos are able to successfully complete most tasks.

**Generalization.** We find that internet-scale pretraining enables UniPi to generalize to novel task commands and scenes in the test split not seen during training, whereas UniPi trained only on task-specific robot data fails to generalize. Specifically, Figure 8 shows the results of novel task commands that do not exist in the Bridge dataset. Additionally, UniPi is relatively robust to background changes such as black cropping or the addition of photo-shopped objects as shown in Figure 9.

# 5 Related Work

**Learning Generative Models of the World.** Models trained to generate environment rewards and dynamics that can serve as "world models" for model-based reinforcement learning and planning have been recently scaled to large-scale architectures developed for vision and language [9, 25, 30, 31]. These works separate learning the world model from planning and policy learning, and arguably present a mismatch between the generative modeling objective of the world and learning optimal policies. Additionally, learning a world model requires the training data to be in a strict state-action-reward format, which is incompatible with the largely available datasets on the internet, such as YouTube videos. While methods such as VPT [32] can utilize internet-scale data through learning an inverse dynamics model to label unlabled videos, an inverse dynamics model itself does not support model-based planning or reinforcement learning to further improve learned policies beyond imitation learning. Our text-conditioned video policies can be seen as jointly learning the world model and conducting hierarchical planning simultaneously, and is able to leverage widely available datasets that are not specifically designed for sequential decision making.

**Diffusion Models for Decision Making.** Diffusion models have recently been applied to different decision making problems [21, 27, 33, 34, 35, 36]. Most similar to this work, [21] trained an unconditional diffusion model to generate trajectories consisting of joint-based states and actions, and used a separately trained reward model to select generated plans. On the other hand, [27] trained a conditional diffusion model to guide behavior synthesis from desired rewards, constraints or agent skills. Unlike both works, which learn task-specific policies from scratch, our approach of text-condition video generation as a universal policy can leverage internet-scale knowledge to learn generalist agents that can be deployed to a variety of novel tasks and environments. Additionally, [37]

applied web-scale text-conditioned image diffusion to generate a goal image to condition a policy on, whereas our work uses video diffusion to learning universal policies directly.

**Learning Generalist Agents.**    Inspired by the success of large-scale pretraining in vision and language domains, large-scale sequence and image models have recently been applied to learning generalist decision making agents [6, 26, 38]. However, these generalist agents can only operate under environments with the same state and action spaces (e.g., Atari games) [26, 38], or require studious tokenization [6] that might seem unnatural in scenarios where different environments have distinct state and actions spaces. Another downside of using customized tokens for control is the inability to directly utilize knowledge from pretrained vision and language models. Our approach, on the other hand, uses text and images as universal interfaces for policy learning so that the knowledge from pretrained vision and language models can be preserved. The choice of diffusion as opposed to autoregressive sequence modeling also enables long-term and hierarchical planning.

**Learning Text-Conditioned Policies.**    There has been a growing amount of work using text commands as a way to learn multi-task and generalist control policies [39, 40, 24, 41, 42, 43]. Different from our framing of video-as-policies, existing work directly trains a language-conditioned control policy in the action space of some specific robot, leaving cross-morphology multi-environment learning of generalist agents as an unsolved problem. We believe this paper is the first to propose images as a universal state and action space to enable broad knowledge transfer across environments, tasks, and even between humans and robots.

## 6    Conclusion

We have demonstrated the utility of representing policies using text-conditioned video generation, showing that this enables effective combinatorial generalization, multi-task learning, and real world transfer. These positive results point to the broader direction of using generative models and the wealth of data on the internet as powerful tools to generate general-purpose decision making systems.

**Limitations.**    Our current approach has several limitations. First, the underlying video diffusion process can be slow, it can take a minute to generate highly photorealistic videos. This slowness can be overcome by distilling the diffusion process into a faster sampling network [44], which in our initial experimentation resulted in a 16x speed-up. UniPi may further be sped up with by faster speed samplers in diffusion models. Second, the environments considered in this work are generally fully observed. In partially observable environments, video diffusion models might make hallucination of objects or movements that are unfaithful or not in the physical world. Integrating video models with semantic knowledge about the world may help resolve this issue, and the integration of UniPi with LLMs would be an interesting direction of future work.

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

# A  Architecture, Training, and Evaluation Details

## A.1  Video Diffusion Training Details

We use the same base architecture and training setup as [45] which utilizes Video U-Net architecture with 3 residual blocks of 512 base channels and channel multiplier [1, 2, 4], attention resolutions [6, 12, 24], attention head dimension 64, and conditioning embedding dimension 1024. We use noise schedule log SNR with range [-20, 20]. We make modifications Video U-Net to support first-frame conditioning during training. Specifically, we replicate the first frame to be conditioned on at all future frame indices, and apply temporal super resolution model condition on the replicated first frame by concatenating the first frame channel-wise to the noisy data similar to [46]. We use temporal convolutions as opposed to temporal attention to mix frames across time, to maintain local temporal consistency across time, which has also been previously noted in [19]. We train each of our video diffusion models for 2M steps using batch size 2048 with learning rate 1e-4 and 10k linear warmup steps. We use 256 TPU-v4 chips for our first-frame conditioned generation model and temporal super resolution model.

We use T5-XXL [22] to process input prompts which consists of 4.6 billion parameters. For combinatorial and multi-task generalization experiments on simulated robotic manipulation, we train a first-frame conditioned video diffusion models on 10x48x64 videos (skipping every 8 frames) with 1.7B parameters and a temporal super resolution of 20x48x64 (skipping every 4 frames) with 1.7B parameters. The resolution of the videos are chosen so that the objects being manipulated (e.g., blocks being moved around) are clearly visible in the video. For the real world video results, we finetune the 16x40x24 (1.7B), 32x40x24 (1.7B), 32x80x48 (1.4B), and 32x320x192 (1.2B) temporal super resolution models pretrained on the data used by [19].

## A.2  Inverse Dynamics Training Details

UniPi's inverse dynamics model is trained to directly predict the 7-dimensional controls of the simulated robot arm from an image observation mean squared error. The inverse dynamics model consists of a 3x3 convolutional layer, 3 layers of 3x3 convolutions with residual connection, a mean-pooling layer across all pixel locations, and an MLP layer of (128, 7) channels to predict the final controls. The inverse dynamics model is trained using the Adam optimizer with gradient norm clipped at 1 and learning rate 1e-4 for a total of 2M steps where linear warmup is applied to the first 10k steps.

## A.3  Baselines Training Details

We describe the architecture details of various baselines below. The training details (e.g., learning rate, warm up, gradient clip) of each baseline follow those of the inverse dynamics model detailed above.

**Transformer BC [6, 26].**  We employ the same transformer architecture as the 10M model of [26] with 4 attention layers of 8 heads each and hidden size 512. We apply 4 layers of 3x3 convolution with residual connection to extract image features, which, together with T5 text embeddings, are used as inputs to the transformer. We additionally experimented with vision transformer style linearization of the image patches similar to [26], but found the performance to be similar. We use a context length of 4 and skip every 4 frames similar to UniPi's inverse dynamics. We tried increasing the context length of the transformer to 8 but it did not help improve performance.

**Transformer TT [25].**  We use a similar transformer architecture as the Transformer BC baseline detailed above. Instead of predicting the immediate next control in the sequence as in Transformer BC, we predict the next 8 controls (skipping every 4 controls similar to other baselines) at the output layer. We have also tried autoregressively predicting the next 8 controls, but found the errors to accumulate quickly without additional discretization.

**State-Based Diffusion [21].**  For the state-based diffusion baseline, we use a similar architecture as UniPi's first-frame conditioned video diffusion, where instead of diffusing and generating future

image frames, we replicate future controls across different pixel locations and apply the same U-Net structure as UniPi to learn state-based diffusion models.

## A.4 Details of the Combinatorial Planning Task

In the combinatorial planning tasks, we sample random 6 DOF poses for blocks, colored bowls, the final placement box. Blocks start off uncolored (white) and must be placed in a bowl to obtain a color. The robot then must manipulate and move the colored block to have the desired geometric relation in the placement box. The underlying action space of the agent corresponds to 6 joint values of robot plus a discrete contact action. When the contact action is active, the nearest block on the table is attached to the robot gripper (where for methods that predict continuous actions, we thresholded action prediction $> 0.5$ to correspond to contact). Given individual action predictions for different models, we simulate the next state of the environment by running the joint controller in Pybullet to try reach the predicted joint state (with a timeout of 2 seconds due to certain actions being physically infeasible). As only a subset of the video dataset contained action annotations, we trained the inverse-dynamics model on action annotations from 20k generated videos.

## A.5 Details of the CLIPort Multi-Environment Task

In the CLIPort environment, we use the same action space as the combinatorial planning tasks and execute actions similarly using the built in joint controller in Pybullet. As our training data, we use a scripted agent on `put-block-in-bowl-unseen-colors`, `packing-unseen-google-objects-seq`, `assembling-kits-seq-unseen-colors`, `stack-block-pyramid-seq-seen-colors`, `tower-of-hanoi-seq-seen-colors`, `assembling-kits-seq-seen-colors`, `tower-of-hanoi-seq-unseen-colors`, `stack-block-pyramid-seq-unseen-colors`, `packing-seen-google-objects-seq`, `packing-boxes-pairs-seen-colors`, `packing-seen-google-objects-group`. As our test data, we used the environments `put-block-in-bowl-seen-colors`, `packing-unseen-google-objects-group`, `packing-boxes-pairs-unseen-colors`. We trained the inverse dynamics on action annotation across the 200k generated videos.

# B Additional Results

## B.1 Additional Results on Combinatorial Generalization

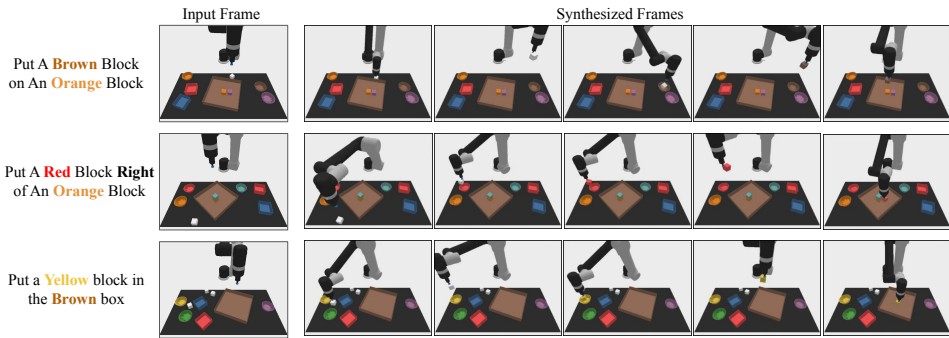

Figure 10: **Combinatorial Video Generation.** Additional results on UniPi's generated videos for unseen language goals at test time.

## B.2 Additional Results on Multi-Environment Transfer

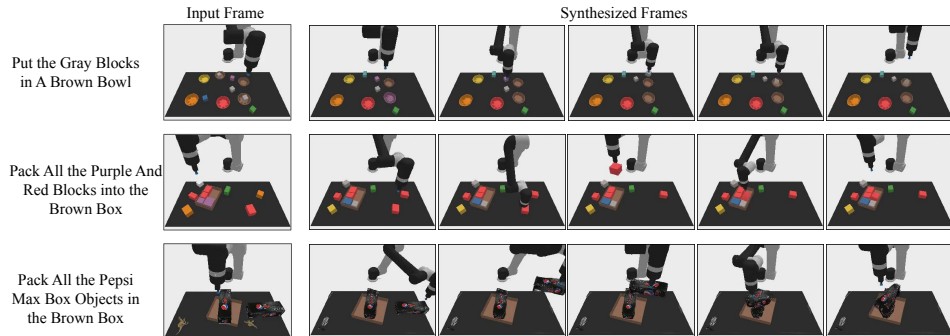

Figure 11: **Multitask Video Generation.** Additional results on UniPi's generated video plans on different new tasks in the multitask setting.

## B.3 Additional Results on Real-World Transfer

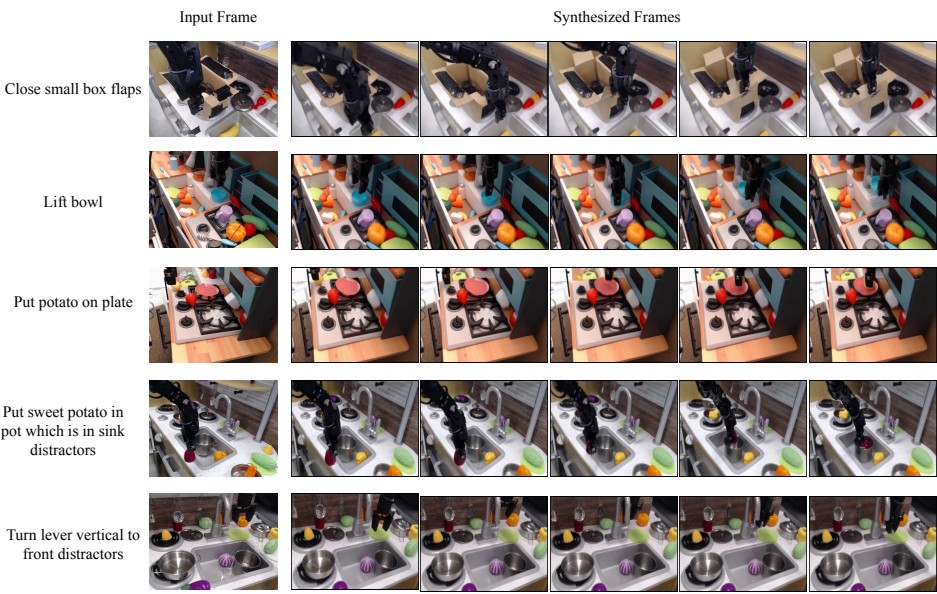

Figure 12: **High Fidelity Plan Generation.** Additional results on UniPi's high resolution video plans across different language prompts.

