# OpenReview forum: "Learning Universal Policies via Text-Guided Video Generation"
_NeurIPS.cc/2023/Conference — NeurIPS 2023 spotlight_

### Official Review · Reviewer_5Y9t · 2023-06-19

**Soundness:** 3 good
**Presentation:** 3 good
**Contribution:** 3 good
**Rating:** 6
**Confidence:** 4

**Summary:**

This paper introduces a novel approach to text-conditioned video generation task  by treating it as a goal-conditioned RL, where the text is formulated as the goal and the multi-step image sequences in the video are the consecutive observations. The proposed method, termed Unified Predictive Decision Process (UPDP), aims to learn effective control strategies. Compared to traditional MDPs, the sequences in UPDP are determined by both the initial frame and the task description. This enables bypassing reward design and facilitates non-Markovian modeling. To model decision-making within UPDP, this paper introduces UniPi, which consists of two key components: i) a diffusion model planner and ii) a task-specific action generator. Empirical evaluations on various control tasks demonstrate the effectiveness of the proposed method.

**Strengths:**

1. This paper is clear to follow.
2. The proposed method is effective and seems reasonable.
3. A large number of experiments showing performance gain.
4. Synthesized Frames are illustrative and interesting.

**Weaknesses:**

1. While the high-level idea presented in this paper is intuitive and reasonable, the detailed descriptions of certain specific modules require further emphasis. For instance, the video-based planner $\rho(\cdot|x_0,c)$ is employed to generate image sequences (or videos). However, the role of the action in this context is not clearly defined. If the planner is unrelated to the action, it raises questions regarding how the action can influence the state. From the statement in lines 115-116 showing that "This design choice isolates planning decisions from action-specific mechanisms, allowing the planner to be environment and agent agnostic", it seems suggest that this module is indeed a forecaster rather than a planner.

2. Text-conditioned generation has gained recent popularity. Although the paper states that planning through video generation poses challenges due to the need for specific initial images to complete tasks, guided generation (conditioned on an initial image and a language description) is a common choice that has been extensively studied. Similarly, inverse dynamics is also a well-explored method that has been proven effective. Therefore, the combination of these techniques may not provide significant novelty from a model design perspective.

3. Some details in the paper may lead to misunderstandings. For example, in section 3.1, the author proposes tiling as a means to ensure trajectory consistency. However, two questions arise: i) Does tiling involve concatenating the generated frame with the initial image of the video or the ground-truth image at a specific timestep? ii) How exactly does tiling provide trajectory consistency? I recommend the authors extend this paragraph with more details.

4. Figure 2 appears somewhat irregular, as the widths of the noises applied to video diffusion are equal, whereas the noises applied to Temporal Super Resolution vary in width. Furthermore, the outputs of the video diffusion module also exhibit different widths. Ensuring consistency in the widths of these elements would enhance the clarity of the figure.

5. The disentanglement between the planner module and the policy network is presented as a strength of the model. However, it remains unclear how the policy network can impact the planner or interact with the environment in such cases. Section 3.2 attempts to explain this relationship, but the details provided are not easy to follow. It is hypothesized that the video planner solely provides synthesized frames for reference (denoted as $\bar{o}$). However, it is unclear whether, when the environment provides an observation $o$, the inputs for inverse dynamics should be $(o_t, \bar{o}_{t+1})$ or $(\bar{o}\_t, \bar{o}\_{t+1})$. Further clarification is needed in this regard.



**Questions:**

1. The specific difference between the super-resolution and low-temporal-resolution architectures mentioned in lines 179-181 appears to be that the former employs the observation tiled as a condition, while the latter does not. It is unclear whether the super-resolution and low-temporal-resolution architectures refer to the hierarchical structure mentioned in lines 184-194. To enhance clarity, I suggest the authors restructure lines 174-183 accordingly. Alternatively, if these architectures are not the hierarchical structure, the authors should provide a clear explanation or visual representation, such as a model figure, to elucidate the architectures for improved readability.

2. In Appendix A.3, it is unclear whether "replicate future controls across different pixel locations" serves the purpose of concatenating state-action pairs. Further clarification is needed to understand the specific role and functionality of this mechanism.

The other questions are above.



**Limitations:**

As the authors showed, the video diffusion process is very slow, even with some fast sampling methods. It's not suitable for real-time robotic control for now, while might be an interesting domain to explore in future work.

---

> ### Author Rebuttal · Authors · 2023-08-10
>
> Thank you for your detailed comments – please see our response below. Feel free to let us know if you have additional questions or comments.
>
> > Detailed Description of Modules.
>
> We will clarify the detailed description of the modules. Our video planner seeks to construct a sequence of states that represent a plan going from our original start state to the desired final goal state. Since the planner only has to generate a set of feasible states transitions to go from the start to goal state, it does not need to worry about the precise actions to generate to reach each feasible state, which can environment specific (i.e. some environments require you to execute a sequence of actions to open a door while other let you go directly through the door).
>
> > Novelty of Text-Conditioned Generation.
>
> Most existing works in text-to-video generation have focused primarily on their application to AI-CG and most large scale text-to-video models (Phenalki, Make-A-Video, Imagen-Video) typically only generate frames conditioned on natural language. The primary novelty of our approach is formulating the problem of acting as a text-to-video generation problem, where given a language description of goal, we synthesize a set of frames of video denoting how we will act. This further requires some domain specific designs, such as how to condition video generation on observed images, where we propose trajectory observation tiling.
>
> > Tiling to Ensure Trajectory Consistency.
>
> The tiling referred to in this section corresponds to tiling the first observed image to each generated frame in the video. This operation ensures that the background details in the first observed image are more readily captured across each frame of the synthesized video as when denoising each frame in the video, the full conv architecture sees (starting with the first convolution) the direct concatenation of an intermediate noisy video with the first observed frame of the video. We will clarify this in the paper.
>
> > Figure 2.
>
> We have attached a modified version of Figure 2 with equal width in image outputs to the main rebuttal response PDF.  Feel free to let us know any other modifications we can do on the figure to improve clarity.
>
> > Clarification on Inverse Dynamics.
>
> To infer an action using the inverse dynamics model, we take as input $o_t, \overline{o}_{t+1}$, the current observed frame and the next synthesized frame. We will clarify this in the paper.
>
> > L179-L181.
>
> The trajectory consistency through observation tiling section refers to the base video diffusion model in Figure 2, where we want to synthesize a low resolution video given the observed image and text-instruction. The temporal super-resolution model we refer to is not related to the one discussed between L184-L194, but rather the temporal super-resolution architecture from Imagen Video. We will clarify this reference in the main paper.
>
> > Baseline Details.
>
> In our state based diffusion model, we want to diffuse a sequence of states to accomplish the instructed language task. To apply our video-diffusion model architecture to this setting, we take each D dimensional state and tile it to form a HxWxD image. We then train a video diffusion to diffusion on this “state” video and use this to regress our future states to execute. We will include these details in the paper.

---

> > ### Comment · Reviewer_5Y9t · 2023-08-12
> > **Thanks for the rebuttal**
> >
> > I appreciate the author's rebuttal, which addresses some of my concerns/confusion. Yet, I still think the novelty is not strong enough to have an "accept". In sum, I will raise the score to "6".

---

### Official Review · Reviewer_VoKS · 2023-07-04

**Soundness:** 4 excellent
**Presentation:** 4 excellent
**Contribution:** 3 good
**Rating:** 6
**Confidence:** 5

**Summary:**

This paper frames the sequential decision-making problem as a text-conditioned video generation problem. Given a text-encoded specification of a desired goal and the first frame with the initial configuration, a planner generates a set of future frames that depict planned actions. The generated video is then used to extract control actions. This framework facilitates leveraging pretrained language embeddings and large-scale internet videos to enable combinatorial generalization and knowledge transfer across diverse tasks.

**Strengths:**

1) Casting sequential decision-making as an synthesis problem naturally allows us to leverage a growing wealth of existing research in large-scale image/video/language generative models that encapsulate valuable world models for robotics.

2) Strong experiments: both simulation and real-world tasks are included.



**Weaknesses:**

1) Some overlap with prior work: the bulk of the framework seems to borrow heavily from Decision Diffuser[1]. Specifically, [1] also uses classifier-free guidance diffusion to generate a sequence of states, followed by some inverse dynamics modeling to identify the action to execute. The main innovations of the work are centered around substituting engineering components in prior works; e.g. replacing unconditional generation from something like Diffuser[2] with conditional generation

[1] [Is Conditional Generative Modeling all you need for Decision Making?](https://arxiv.org/abs/2211.15657)

[2] [Planning with Diffusion for Flexible Behavior Synthesis](https://arxiv.org/abs/2205.09991)

**Questions:**

1) I wonder how much of the combinatorial generalization exhibited by the model is dependent on the quality of the text embedding. For example, as shown in [1], using something like CLIP language embeddings can make it difficult to capture more abstract notions like "left" or "right" relative spatial positioning, affecting combinatorial generalization in some of the tasks shown in Section 4.1. Could the authors include an sensitivity analysis on the T5 embeddings?

2) The authors claim this approach allows for more universality when it comes to environment diversity or reward specification. But it seems to me that the inverse dynamics model discussed in Section 3.2 would have to be retrained for different tasks anyways. How do the parameter/data scales for this (smaller) model compare to the diffusion component?

3) How does performance for combinatorial generalization in Section 4.1 scale compared to something like BC? i.e. is there is a ceiling to training inverse dynamics on generative data -- could the authors demonstrate how performance scales with number of examples compared to something like [1] or [2]?

[1] [Programmatically Grounded, Compositionally Generalizable Robotic Manipulation](https://arxiv.org/abs/2304.13826)

[2] [CLIPort: What and Where Pathways for Robotic Manipulation](https://arxiv.org/abs/2109.12098)

**Limitations:**

Yes, the authors have discussed some limitations in their concluding remarks.

---

> ### Author Rebuttal · Authors · 2023-08-10
>
> Thank you for your detailed comments – please see our response below. Feel free to let us know if you have additional questions or comments.
>
> > Overlap with Prior Work.
>
> We believe the primary novelty of our work over past work such as Decision Diffuser is the construction of a large-scale model for decision making that can be directly learned from Internet data. While Decision Diffuser, similar to our work, uses diffusion models, in combination with classifier free guidance, to predict states and actions, it assumes the presence of existing datasets of states and actions in a domain of interest. In contrast, our work is able to use the existing video information on the internet to learn to make decisions, by casting the decision making problem is that of generating a video of actions to execute conditioned on a text description of the actions you wish to take.
>
> > Effect of Language Embedding on Combinatorial Generalization.
>
> We ran an ablation experiment where we tested the combinatorial generalization ability of our approach with different sizes of language embeddings from T5. While we found that different sizes of language embeddings did really affect the video quality, we indeed found that larger language embedding sizes substantially improved combinatorial generalization (as measured by CLIP score on Bridge).
>
> | | CLIP| FID|
> | ---- | ---- | ---|
> |T5_small  |   22.75  |    15.53  |
> |T5_large  |   22.78   |   16.07   |
> |T5_XXL   |   24.54   |   14.54   |
> |T5_XXL (No Pretrain)   |   24.43  |   17.75   |
>
>
> > Inverse Dynamics Model.
>
> The inverse dynamics model is very small (<1M parameters) and trained on substantially less resources (1 TPU for 12 hours)  while the video model which is large (5B parameters) and trained on substantially more resources (256 TPUs for several days). While the inverse dynamics would need to be trained per environment, the model only needs to learn to predict the action that can transition between two frames which is substantially easier to learn.
>
> > Scaling of Combinatorial Generalization.
>
> In principle, we believe that both our text-conditioned video generation approach and language conditioned BC would scale well with increased amounts of data. However, while there are plentiful amounts of videos on the internet (and many commercial large text-to-video models like Gen2 that demonstrate very good combinatorial generalization) there is much less labeled action data, making it much easier to get a combinatorially generalized version of our approach.
>
> We don’t believe there is a limit to training inverse dynamics models on generated data and believe it's actually much easier to train an inverse dynamics model than a BC policy that generalizes well as also theoretically shown in [1]. This is because the inverse dynamics model only has a task to  infer an action given both present and future states while a policy must anticipate the next action to take to maximize reward across all future states [1]. If the reviewer desires, we are happy to add an additional experiment demonstrating scaling of the performance of the inverse dynamics model with number examples in the final version of the paper.
>
> [1] Brandfonbrener et al. Inverse Dynamics Pretraining Learns Good Representations for Multitask Imitation

---

> > ### Comment · Reviewer_VoKS · 2023-08-15
> >
> > I thank the authors for their response. I have two follow-up questions/comments:
> >
> > 1) My concern with the effect of language embedding on combinatorial generalization is that such generalization is dependent *entirely* on using very strong text embeddings. That is, below some certain representational capacity for said embeddings, combinatorial generalization disappears entirely. This is important to me because it helps weigh the contributions of the rest of the framework (e.g. the diffusion approach to decision making), against "just using better text representations." Thus, I feel that reporting results on different sizes of T5 is not really addressing this concern, because the scale of the text data (and thus the upper bound to the semantics represented by such embeddings), is the same. What I would prefer to see is using both weaker and stronger text embeddings that are completely orthogonal to T5 (e.g. older CLIP embeddings using as in ProgramPort or CLIPort, or some of the newer open LLM embeddings).
> >
> > 2) I am familiar with the work of Brandfonbrener et al., but it's not clear to me their findings/analysis extend to the domain of learning on generated data. Intuitively, I agree that it makes sense that learning inverse dynamics is going to be more sample-efficient than BC here as well, but if models learned via both approaches collapse after some N samples, for an uninterestingly low N (e.g. comparable to existing real-world datasets upon which one can reasonably do BC), then that for me nullifies some core advantages of the proposed approach, even if it does marginally better than traditional BC. It would still be great if the authors could include in their revised/final draft this scaling experiment.
> >
> > I would also like to hear from my fellow reviewers; in the meantime, my rating stands.

---

> > > ### Author Response · Authors · 2023-08-16
> > > **Reply to Reviewer VoKS**
> > >
> > > Thank you for your comments – please see our clarifications below:
> > >
> > > > Benefit of text embeddings
> > >
> > > We would like to clarify why we conducted the ablation on different size of T5 embeddings. We wanted to demonstrate that less powerful text embeddings (e.g., those from T5-small) can still results in successful plans despite T5-small having substantially lower language modeling and compositional performance in the original T5 paper [1]. UniPi is able to utilize T5-small’s embedding to generate successful plans on new prompts, which implies that quality of language embedding is not essential for successful plan extraction.
> > >
> > > A variety of prior works have demonstrated the efficacy of using CLIP embeddings in diffusion models. For instance the DALLE-2 model is based off the CLIP text encoder, but is able to demonstrate good combinatorial generalization across different language prompts (including simple relations between objects). An analysis of the effect of text-embeddings on text-to-image generation is studied in [2] (Figure A.5) and its found that T5-Small (60M), T5-Large (770M) (which are substantially smaller embeddings than T5-XXL (12B))  are substantially worse than CLIP in image generation metrics, suggesting that in our above analysis, CLIP embeddings would likely also successfully generate plans.
> > >
> > > It’s difficult for us to directly evaluate the effect of using CLIP embeddings in our text-to-video setup, as our existing codebase and pretrained models are based on T5 embeddings, which have a different shape that of CLIP.
> > >
> > > > Inverse Dynamics Analysis
> > >
> > > We would like to clarify that the inverse dynamics and behavioral cloning models are directly trained on real data, as opposed to generated data, so the work of Brandfonbrener et al. would apply in our setting. We take the inverse dynamics model learned on real data and apply it to generated frames, which we find to be effective, as the generative model is directly fitting the distribution of real data (and would perfectly fit it in the theoretical limit).
> > >
> > > [1] Exploring the Limits of Transfer Learning with a Unified Text-to-Text Transformer. JMLR 2020
> > >
> > > [2] Photorealistic Text-to-Image Diffusion Models with Deep Language Understanding. NeurIPS 2022

---

### Official Review · Reviewer_eKLN · 2023-07-07

**Soundness:** 3 good
**Presentation:** 3 good
**Contribution:** 3 good
**Rating:** 7
**Confidence:** 3

**Summary:**

The authors utilize the enhanced capabilities of text-guided image synthesis, a recent advancement in deep learning, to engineer general-purpose embodied agents capable of sequential decision-making.

The proposed method involves using language instructions as inputs to a text-conditioned video generation model, specifically video diffusion. Actions are then inferred from the generated video using a learned inverse dynamics model.

As agent learning from video play has been a well-established approach in reinforcement learning and imitation learning for complex tasks, translating language instructions into video for general-purpose agents can be viewed as a universally applicable strategy. This approach addresses the challenges of diversity and heterogeneity in agent environments, offering a broader, more versatile solution that can leverage also pretrained large-scale language and language-video models.

**Strengths:**

This paper demonstrates how to harness the potential of text-guided image synthesis for agent learning, providing evaluations through multiple experimental scenarios.

The paper is insightful, and it can offer several meaningful perspectives for readers interested in using LLMs and multi-modal pretrained models for agent learning.

**Weaknesses:**

The proposed method in this paper can be seen novel, marking its originality, and showcasing insights on the utilization of multi-modal models.  However, I also think that the method description in Section 3.1 falls short on illustrating clear technical contributions. It seems to only incorporate the existing diffusion model and the inverse dynamics model in the proposed framework.

Despite the innovative nature of the broader concept, a more detailed explanation and comprehensive analysis regarding how to improve and generalize the proposed method can enhance the clarity and overall influence of the paper; e.g., comparing with language instruction-following agents with multi-modal capabilities, and comparing with skill-based hierarchical RL approach that generate latent skill sequences (that can be also translated into actions later).


**Questions:**

Could the authors explain the benefits of the proposed method, specifically in comparison to other instruction-following agents that can use pretrained multi-modal models?

If each video frame generated can be seen as sub-goal representations in the task, and each sub-goal is dealt by the inverse dynamics model, hierarchical planning in Section 3.1 seems to be critical for the effective handling of long-horizon tasks. Could the authors more describe the procedure of sampling videos?

In line 215, the open-loop controller is chosen for computation efficiency. Could the author explain more? What if the stochastic environment has some randomness?

Could the authors explain why Transformer-BC and TT achieve low performance in the experiments?

In Figure 5, the adaptable planning scenario is not clear.


**Limitations:**

The conclusion has specified the limitations including computation load of video diffusion.

---

> ### Author Rebuttal · Authors · 2023-08-10
>
> Thank you for the positive feedback on our work! We address your questions as follows.
>
> > Highlighting contribution in method section
>
> We will update our method sections to highlight the contributions, which includes (1) how to re-purpose text-to-video models designed for media and entertainment to be a useful tool for control through frame-conditioning, (2) how to overcome difficulties around generating consistent frames across time, and (3) how to conduct hierarchical planning effectively. We believe that these contributions indeed address challenging problems in both video diffusion models and planning and control.
>
> > Procedure of sampling videos to address temporal hierarchy and long-horizon planning
>
> Generating videos hierarchically at the right granularity across time is indeed important to ensure UniPi’s performance. To sample a video plan in the simulated robotics experiments, we first sample 10 frames with a larger frame-skip between each frame (8). Then conditioned on these 10 frames, we fill in the intermediate frames, resulting in 20 frames with frame-skip 4. We found frame-skip 4 results in the right granularity for training an effective inverse dynamics model.
>
> > Advantage over language instruction following agent pretrained on multimodal data
>
> Most instruction follow agents using multimodal data are typically initialized from pretrained VLM models. These models are typically trained with captioned image and text-pairs, but typically do not contain much motion / physics information about the environment. In contrast, in our approach, we can train on a wealth of existing language annotated video data, which captures much more information about how to act in environments and the physics of the environments. This allows our approach to transfer world-knowledge about how to do particular tasks, such as precise visual motions to open a door handle which are not available in pretrained VLM models.
>
>
> > Open loop control
>
> A limitation of UniPi is the high computational cost of video diffusion to generate a video plan, making it costly to perform open loop control. In stochastic environments, we could choose to regenerate video plans in the case in which observations do not match the ones in our plan,  or wecan batch sample video plans ahead of time switch between generated plans to address changes in the environment. There is also a wealth of work on improving the video sampling speed of diffusion models.
>
> > Poor performance of Transformer-BC and TT
>
> We believe the poor performance of Transformer-BC and TT is due to the long task horizon of demonstrations and the inability of these agents to accurately synthesize and follow the long horizon plans in this environment. As the different steps in the environment are executed, errors accumulate in the observation space and the agents fail to correctly finish the task.
>
> > Figure 5 adaptable planning clarification
>
> Figure 5 shows that in addition to using text to guide plan generation, UniPi can further utilize intermediate frames by fixing a particular future frame during sampling to guide generated plans towards moving either the left or the right block. We have updated the caption to make this clear. Please let us know if you have additional confusion.

---

> > ### Comment · Reviewer_eKLN · 2023-08-20
> >
> > I'd like to extend my thanks for the comprehensive response, which addresses most of the concerns I raised. I have decided to maintain my original score of accept.

---

### Official Review · Reviewer_voju · 2023-07-07

**Soundness:** 3 good
**Presentation:** 4 excellent
**Contribution:** 3 good
**Rating:** 7
**Confidence:** 4

**Summary:**

A general framework, the Unified Predictive Decision Process (UPDP), was proposed in this paper. It leverages images as a universal interface, texts as reward specifiers and an independent planning module for policy synthesis. Powerful diffusion model was adopted into the framework of UPDP to generate authentic future video frames and inverse dynamics was used to generate downstream policies.

**Strengths:**

1. This paper is well written, clear and easy to understand.
2. A very intriguing introduction of UPDP framework. UPDP enables better utilization and knowledge transfer of large-scale generative pretraining which may demonstrate a broader impact in the future applications.
3. Extensive experiments demonstrate the effectiveness of the proposed method, including high-quality video generation,  combinatorial generalization and multi-environment transfer.

**Weaknesses:**

1: Diffusion model: The adapation of diffusion model to UPDP lacks novelty. Temporal superresolution and tiling the context frame are commonly used tricks in video diffusion model. Although this can be regarded as "simple yet effective", this paper can be further improved if authors can include more domain specific design into the instantiation of UPDP with diffusion models.

2. Metrics: For table 4, first there is a typo in line 302 saying "higher FID and FVD" while actually it's lower FID and FVD. Besides, FID and FVD are also not informative in this case because they only measure the distance between the distribution of generation and real data, which cannot tell if the model follow the text prompt correctly. Beyond training a classifier, a better metric could be measuring IoU of bounding boxes between groundtruth and generations, which are produced by a pretrained object detector, or human evaluation.



**Questions:**

1. In Figure 7, are there images generated by the diffusion model or videos taken during the action excution? The reflection on the microoven in "Turn Faucet Left" row looks too consistent to be something generated by diffusion while this reflection is not shown in the input frame.



**Limitations:**

Authors talked about the potential improvement of generation speed for diffusion model part.

In general, I think this is a solid and intriguing work.

---

> ### Author Rebuttal · Authors · 2023-08-10
>
> Thank you for the detailed review. Please find our response below.
>
> > Novelty of diffusion models for UPDP
>
> We agree that frame conditioning and temporal super-resolution are not new in video diffusion models, but adapting them to control and hierarchical planning have not been done before. To our knowledge, UniPi is the first to extend video diffusion to perform planning in the image space, and the implications are significant given the continual development of text-to-video foundation models and the ability to leverage internet-scale video data to improve decision making. Furthermore, we note that the introduction of UPDP as a more practical alternative to the more restrictive MDP is also novel. We will update the manuscript to include these discussions.
>
> > IoU metric for evaluation
>
> Thank you for the suggestion. We will try this evaluation metric for the final version of the manuscript, although we suspect that using IoU of bounding boxes might have more false negatives due to the stochasticity of the generated plans (i.e., there are multiple ways to complete a task and the final frame might look different between the ground truth and generated plans, despite generated plans also complete the task).
>
> > Figure 7 clarification
>
> The videos in Figure 7 are generated by the diffusion model. We have attached a few more generated videos conditioned on the same first frame and noted the generated videos are diverse.
>
> > Typos on line 302
>
> We have updated the manuscript to fix the typo.

---

> > ### Comment · Reviewer_voju · 2023-08-17
> > **Response to rebuttal**
> >
> > All of my concerns are well addressed. Thus, I am pleased to raise my score to accept.

---

### Author Rebuttal · Authors · 2023-08-10

We thank reviewers for their positive reviews and feedback. Reviewers noted that the paper was well written (Reviewer voju, 5Y9t), insightful (Reviewer eKLN), and had strong experiments (Reviewer VoKS). Reviewers voju, VoKS, 5Y9t had some concerns about novelty which we address below.

## Novelty

The main novelty of our work lies in our formulation that enables us to leverage internet video as an effective data modality for decision making. Most recent works on leveraging internet data have used multimodal models, primarily VLMs, are trained on captioned image-text pairs, to construct agents. While such agents will have rich prior knowledge about the underlying semantics of individual images, they lack knowledge about motion of objects or their physics, and for example would not know how visually one should go around opening something like a door handle. In contrast, learning from video allows us to learn the motions of people and the world, enabling us to have prior knowledge about the mechanisms of opening a door, or what types of object dynamics or movement are possible.

To leverage internet video for decision making, we propose the UPDP decision making formulation, which casts decision making in different environments to be very closely related to video generation, enabling us to transfer this wealth of knowledge. While our resultant instantiation of UPDP shows some similarity to Decision Diffuser and existing video models, these are particular instantiations of our approach, and our formulation to could be also applied to newly discover or other existing generative models of video.

---

### Decision · Program_Chairs · 2023-09-21

**Decision:**

Accept (spotlight)

**Comment:**

The paper provides a technique that use generative model with diffusion models to perform planning. The scores from all reviewers are positive and most reviewers find the approach interesting and the results are convincing. The AC agrees with the reviewers on accepting the paper.